# An Arg/Ala-rich helix in the N-terminal region of *M. tuberculosis* FtsQ is a potential membrane anchor of the Z-ring

Sean T. Smrt[1,2,6], Cristian A. Escobar [1,3,6], Souvik Dey [4], Timothy A. Cross[1,2,3✉] & Huan-Xiang Zhou [4,5✉]

*Mtb* infects a quarter of the worldwide population. Most drugs for treating tuberculosis target cell growth and division. With rising drug resistance, it becomes ever more urgent to better understand *Mtb* cell division. This process begins with the formation of the Z-ring via polymerization of FtsZ and anchoring of the Z-ring to the inner membrane. Here we show that the transmembrane protein FtsQ is a potential membrane anchor of the *Mtb* Z-ring. In the otherwise disordered cytoplasmic region of FtsQ, a 29-residue, Arg/Ala-rich α-helix is formed that interacts with upstream acidic residues in solution and with acidic lipids at the membrane surface. This helix also binds to the GTPase domain of FtsZ, with implications for drug binding and Z-ring formation.

[1] National High Magnetic Field Laboratory, Tallahassee, FL 32310, USA. [2] Department of Chemistry and Biochemistry, Florida State University, Tallahassee, FL 32306, USA. [3] Institute of Molecular Biophysics, Florida State University, Tallahassee, FL 32306, USA. [4] Department of Chemistry, University of Illinois Chicago, Chicago, IL 60607, USA. [5] Department of Physics, University of Illinois Chicago, Chicago, IL 60607, USA. [6] These authors contributed equally: Sean T. Smrt, Cristian A. Escobar. ✉email: timothyacross@gmail.com; hzhou43@uic.edu

In *Mycobacterium tuberculosis* (*Mtb*), FtsQ is an essential transmembrane protein that participates in the development of the divisome, the machinery responsible for bacterial cell division[1–3]. The divisome comprises dozens of transmembrane and water-soluble proteins. While its composition can vary from organism to organism, at the heart is FtsZ, a water-soluble protein with a globular GTPase domain and a disordered C-tail; the GTPase domain polymerizes to form the Z-ring and initiate cell division. The Z-ring must be anchored to the inner membrane, via interactions with either transmembrane proteins or membrane-tethered water-soluble proteins. While the *Escherichia coli* version typical of bacterial FtsQ proteins has only a short N-terminal cytoplasmic region (27 residues), the *Mtb* version is 99 residues long, prior to a single transmembrane helix and a 191-residue C-terminal periplasmic region that putatively forms a POTRA (or polypeptide transport associated) domain. Both the cytoplasmic region and periplasmic region are critical for *Mtb* cell length maintenance[3]. Biophysical characterization of the periplasmic region is a subject of debate as the mechanistic attributes of POTRA domains remain to be demonstrated for the *Mtb* FtsQ protein[4]. Here, we focus on the characterization of the *Mtb* FtsQ cytoplasmic region, or FtsQ1-99, and demonstrate its binding intramolecularly in solution, to acidic membranes (mimicking the *Mtb* inner membrane[5]), and to *Mtb* FtsZ.

Membrane anchoring of the Z-ring is well studied in *E. coli* and *Bacillus subtilis*. Two proteins, FtsA and ZipA in *E. coli* and FtsA and SepF in *B. subtilis*, each act as membrane anchors[6–8]. ZipA is a transmembrane protein[9], whereas FtsA and SepF are water-soluble proteins with an amphipathic helix for membrane tethering[10,11]. FtsZ uses its disordered C-tail to bind a structured domain in each of these three proteins for membrane anchoring[11–13]. In *E. coli*, FtsA and ZipA provide redundancy in membrane anchoring, as Z-rings can form in the absence of either one of them but not both[6]. Such redundancy is likewise provided by FtsA and SepF in *B. subtilis*[8]. However, *Mtb* has SepF[14,15] but no FtsA, and may have another unidentified membrane anchor of the Z-ring for redundancy. SepF has been studied as a membrane anchor of the Z-ring in *Corynebacterium glutamicum*[16] and *Mtb*[15,17].

Using several complementary techniques including solution and solid-state NMR spectroscopy and molecular dynamics (MD) simulations, we characterized the conformational and dynamic properties of *Mtb* FtsQ1-99 in solution and its interactions with membranes and *Mtb* FtsZ. The midsection of FtsQ1-99 forms a long α-helix (residues 46–74) that is rich in Arg and Ala residues and interacts with N-terminal acidic residues. This Arg/Ala-rich helix binds to acidic membranes through weak hydrophobic but strong electrostatic interactions. It also binds to the GTPase domain of FtsZ, thereby making FtsQ a potential membrane anchor or a stabilizer of the *Mtb* Z-ring. The binding of FtsQ1-99 to the GTPase domain of FtsZ is in contrast to the latter's interactions with other divisome proteins, which all occur via the disordered C-tail of FtsZ, and has implications for drug binding and Z-ring formation.

## Results

### FtsQ1-99 is largely disordered in solution but forms an Arg/Ala-rich α-helix.

The N-terminal cytoplasmic region of FtsQ, comprising the first 99 residues, is expected to be disordered due to its high content of charged residues: nearly half of the residues are either positively charged (Arg and Lys, 21 total) or negatively charged (Asp and Glu, 20 total) (Fig. 1a). Even more strikingly, the positive and negative charges are separated, with a net negative charge of −14 for the first 44 residues and a net positive charge of +15 for the remaining 55 residues. Consistent with the expectation of disorder, the HSQC spectrum of FtsQ1-99 in solution is well-resolved with a narrow proton chemical shift range (7.8–8.5 ppm) (Fig. 1b). Glu40 is an outlier with a proton chemical shift of 9.1 ppm, due at least in part to its position following a Pro residue.

The full assignments for the backbone N, $H^N$, $C^O$, $C_\alpha$, and $C_\beta$ chemical shifts are presented in Supplementary Table 1. The $C_\alpha$ and $C_\beta$ secondary chemical shifts clearly indicate a long α-helix formed in the midsection of FtsQ1-99, with $\Delta\delta C_\alpha - \Delta\delta C_\beta$ in the range of ~2–4 ppm for residues 50–71 (Fig. 2a). The secondary chemical shifts of the first 37 residues and last 24 residues are

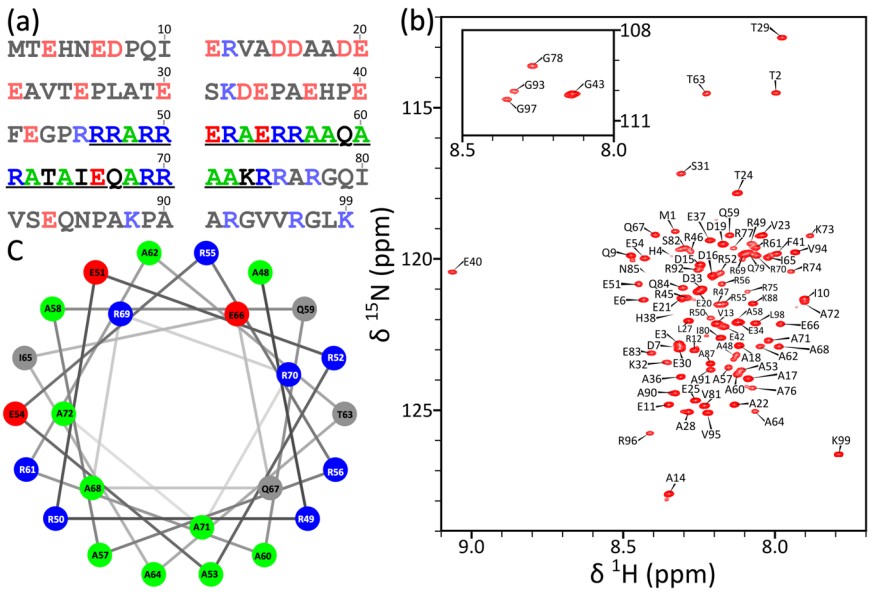

**Fig. 1 Sequence and conformational properties of FtsQ1-99. a** The amino-acid sequence. The helical region is underlined, with Ala residues therein in green; acidic and basic residues are shown as red and blue, respectively, throughout the sequence. **b** ¹H-¹⁵N HSQC spectrum of uniformly ¹³C-¹⁵N-labeled FtsQ1-99 at 370 μM, acquired at 800 MHz and 305 K in 20 mM Tris (pH 6.85) with 50 mM of NaCl. **c** Helical wheel representation for core helical residues 48–72. Acidic, basic, Ala, and other residues are shown in red, blue, green, and grey, respectively. Arg49 and Arg50 define a 100° arc, in which Ala53, Ala57, Ala60, Ala64, and Ala71 fall.

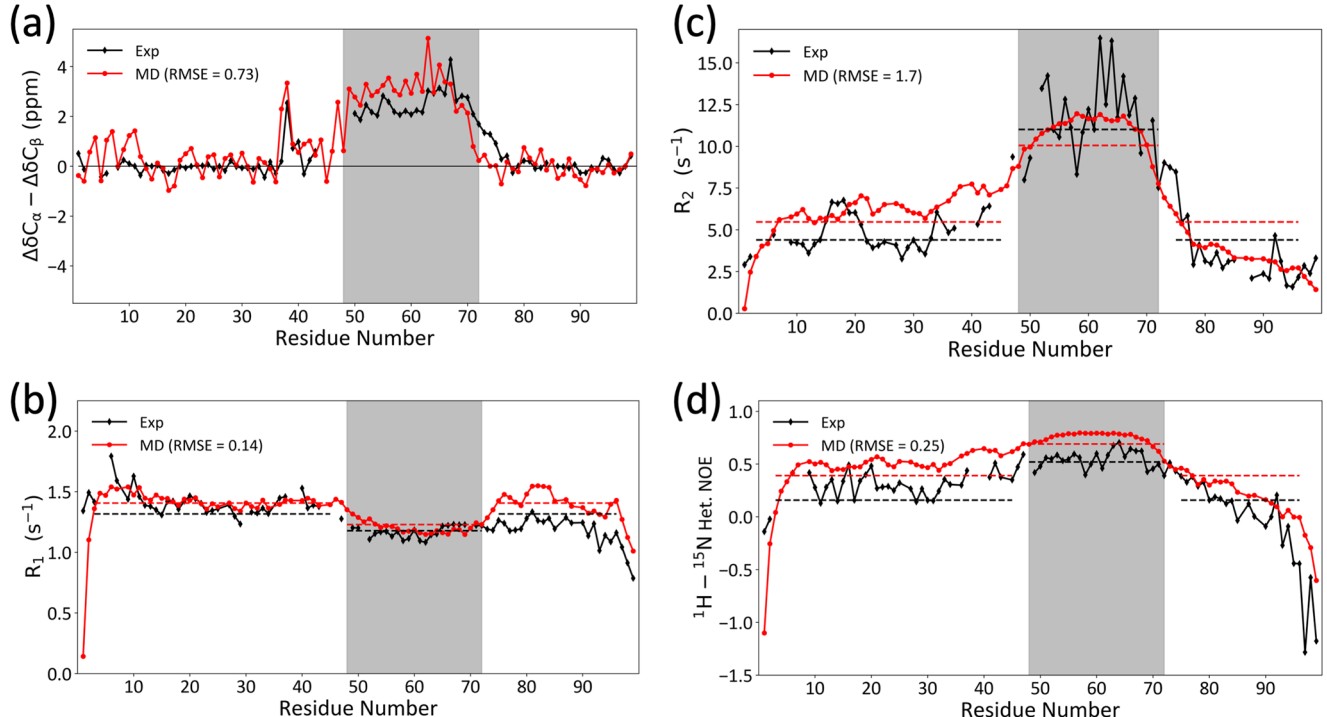

**Fig. 2 Conformational and dynamic properties of FtsQ1-99 determined by solution NMR and MD simulations. a** Secondary chemical shifts ($\Delta\delta C_\alpha$ – $\Delta\delta C_\beta$). For Gly residues, only $\Delta\delta C_\alpha$ are displayed. **b**–**d** $R_2$, $R_1$, and NOE. NMR data are shown in black; MD results are shown in red, with root-mean-square errors (RMSEs) indicated in legends. The core helical region, spanning residues 48 to 72, is shaded. Horizontal dashes indicate average values within the core helical region or non-helical region (residues 3–45 and 75–96).

essentially zero. Backbone NH residual dipolar couplings (RDCs) measured by alignment in stretched polyacrylamide gels delimit this α-helix to be between residues 46 and 74, all of which have RDCs above ~2 Hz (Supplementary Fig. 1a). Some of the peaks could not be assigned because of peak overlap or low peak intensity; this problem is made worse in the RDC experiments because the data analysis requires four separate spectra, and the alignment media increases the rotational correlation time thereby broadening the peaks. Still, the different NMR experiments all point to the helical region being comprised of residues 46–74. Beyond this helical region, RDCs are nearly zero or slightly negative, further documenting disorder in the N-tail (residues 1–45) and C-tail (residues 75–99) of FtsQ1-99. Indeed, RDCs back-calculated from a model where residues 46–74 formed an α-helix agree well with the observed values (Supplementary Fig. 1b, c). This α-helix is quite long, with 29 residues. It is also notable for its composition, with 11 Arg residues, 10 Ala residues, and only 8 other amino acids (including 3 Glu residues). We will therefore refer to this helix as "Arg/Ala-rich". For each type of amino acid, there are significant differences in $C^O$, $C_\alpha$, and $C_\beta$ chemical shifts between helical and nonhelical positions. For reference below, in Supplementary Table 2 we list the chemical shifts of Ala, Arg, and Glu averaged over helical and nonhelical positions.

Additional support for the conclusion that FtsQ1-99 forms an α-helix in an otherwise disordered background is provided by NMR backbone relaxation data (Fig. 2b–d). The transverse ($R_2$) and longitudinal ($R_1$) relaxation rates and the heteronuclear Overhauser effects (NOEs) are relatively uniform for residues 3–45 and 75–96, with average values of 4.4 s$^{-1}$, 1.3 s$^{-1}$, and 0.2, respectively, which are typical of those reported for fully disordered regions[18]. The helical region has elevated $R_2$, depressed $R_1$, and elevated NOEs, similar to relaxation data reported for the C-terminal domain of the Sendai virus nucleoprotein[19], which also features an α-helix in an otherwise

disordered background. The average $R_2$, $R_1$, and NOE values are 11.0 s$^{-1}$, 1.2 s$^{-1}$, and 0.5, respectively, for the helical core (residues 48–72).

We also carried out MD simulations of FtsQ1-99 in solution. Residues 46–74, modeled as helical in the starting structure, stay helical in the simulations (12 replicates, 1134 ns each), except for some fraying at the capping residues (Supplementary Fig. 1d). The N- and C-tails remain disordered, with no stable secondary structures formed. The simulations reproduce well the $C_\alpha$ and $C_\beta$ secondary chemical shifts and the backbone relaxation data (Fig. 2).

**The Arg/Ala-rich α-helix interacts with the N-tail.** The charge separation between the N-tail and the rest of FtsQ1-99 presents possibilities for electrostatic interactions. To investigate these possibilities, we carried out paramagnetic relaxation enhancement (PRE) experiments, in which a single Cys residue was introduced at Thr24, Thr63, or Ala87 for attaching an MTSL spin label (Fig. 3a). These sites were selected to be near the centers of the three sections of FtsQ, namely the N-tail, the Arg/Ala-rich helix, and the C-tail. We also limited to residues that are similar to Cys. The PRE data from the spin labels at positions 24 and 63 demonstrate that the N-tail folds over the helical region. With the spin label at position 24, the entire Arg/Ala-rich helix and a few residues beyond (up to residue 78) experience diminished PREs, reporting the proximity of residue 24 to the helical region in the conformational ensemble of FtsQ1-99. Likewise, the spin label at position 63 suppresses PREs at residues 10–30. However, the C-terminal residues 80–99 are unaffected by either of these spin labels, indicating a lack of interactions between the C-tail and N-tail or Arg/Ala-rich helix. As corroboration, the PRE effects of the spin label at position 87 are entirely confined to the C-tail. The interaction between the Arg/Ala-rich helix and the acidic N-tail is likely important for the stability of the helix.

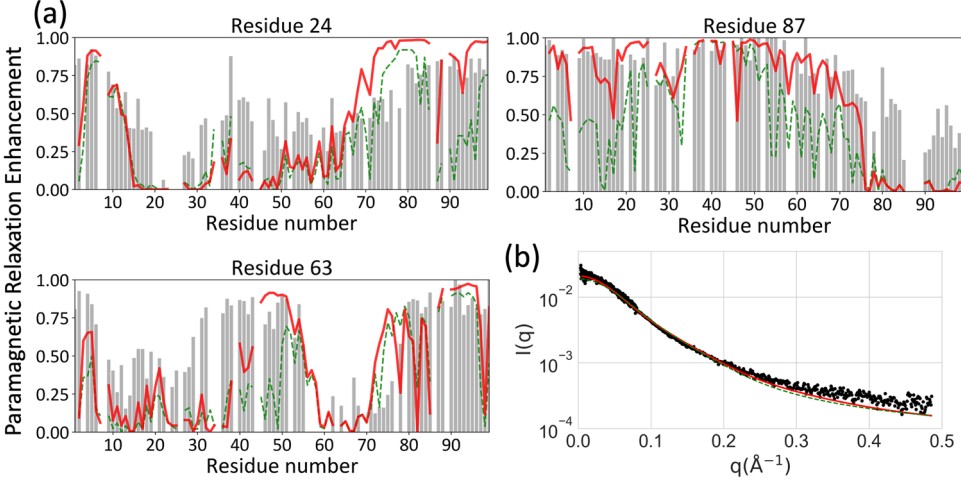

**Fig. 3 Conformational characterization by PRE and SAXS.** Comparison of experimental data and MD results for **a** PRE values with spin labels at residues 24, 63, and 87. **b** SAXS profiles. Experimental data are shown as gray bars or black dots; MD results are shown as green or red curves, calculated over all MD conformations, or excluding those with contacts between residues 1–42 and 73–99.

The MD simulations provide details of the inter-residue interactions in FtsQ1-99 (Supplementary Fig. 2). The sidechain-sidechain contact map shows extensive electrostatic interactions between the N-tail and the Arg/Ala-rich helix. The N-tail acidic residues bridge between the Arg residues, thereby providing both charge neutralization and additional stabilization of the helix (Supplementary Fig. 2, top inset). However, the N-tail also exhibits some tendency to form salt bridges with the C-tail (Supplementary Fig. 2, bottom inset), contradicting the PRE data and revealing limitations of the present force field for intrinsically disordered proteins. When conformations with contacts between residues 1–42 and residues 73–99 are excluded, PRE results calculated from the MD simulations have good agreement with the experimental data (Fig. 3a).

We further characterized the conformational ensemble of FtsQ1-99 by SAXS (Fig. 3b). The SAXS profile is typical of an intrinsically disordered protein. The reproduction by the MD simulations is good, and further improved when conformations with contacts between residues 1–42 and residues 73–99 are excluded.

**The Arg/Ala-rich α-helix binds to acidic membranes.** As shown by the helical wheel projection in Fig. 1c, a subset of the Ala residues in the Arg/Ala-rich helix lines a ridge along a single face of the helix. Due to the amphipathic nature, this helix is expected to bind well with acidic membranes that mimic the inner membrane of *Mtb*[5]. Our pulldown assay indeed shows that FtsQ1-99, while not binding the neutral POPC lipids, binds vesicles when an acidic lipid, POPG, is added to a 40% level (Fig. 4a and Supplementary Fig. 3a). We further confirmed the electrostatic nature of this binding by doing the assay with 8:2 POPG/POPC vesicles at increasing salt concentrations. Strong binding occurs at 50, 125, and 250 mM NaCl, but the bound fraction is reduced significantly at 500 mM NaCl and even further at 1000 mM NaCl. Circular dichroism spectra show that the helical content of FtsQ1-99 is the same whether in solution or bound to 7:3 POPG/POPC vesicles (Supplementary Fig. 3b, c).

To characterize the membrane binding of FtsQ1-99 at the residue level, we acquired the HSQC spectrum of FtsQ1-99 bound to 7:3 POPG/POPC vesicles (Supplementary Fig. 4). The helical region and flanking residues show significant loss in peak intensity due to membrane binding, with resonances for residues 50–80 all but disappearing (Fig. 4b). The loss in peak intensity for

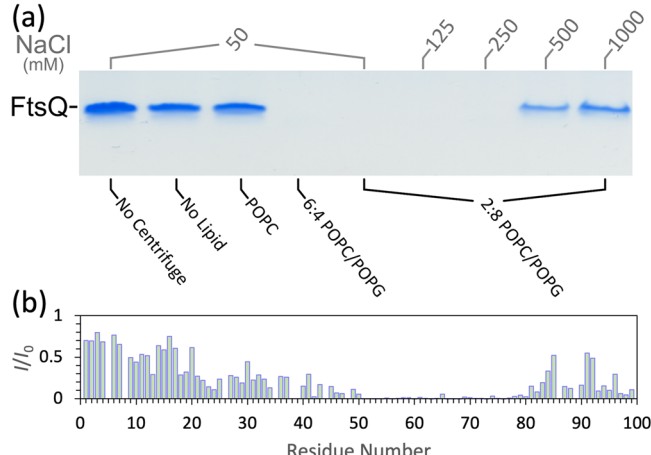

**Fig. 4 Characterization of the binding of FtsQ1-99 with membranes. a** SDS-PAGE gel indicating the fraction of FtsQ1-99 remaining free of lipid vesicles. Gel samples were of supernatant from the indicated mixtures collected immediately after ultracentrifuge condensation of lipid vesicles. All samples contained 320 µM FtsQ1-99 in 20 mM Tris (pH 6.85) in the presence of indicated NaCl and/or lipid vesicles with a final protein-to-lipid ratio of 1:20. **b** Ratio of $^1H$-$^{15}N$ HSQC peak intensities of 130 µM uniformly $^{13}C$-$^{15}N$ labeled FtsQ1-99 in the presence (*I*) and absence (*I*$_0$) of 7:3 POPG/POPC vesicles (1:10 protein-lipid ratio). Acquired at 800 MHz and 305 K in 20 mM Tris buffer (pH 6.85) with 50 mM NaCl.

the helical region (residues 46–74) is expected. The loss experienced by residues 21–45 and 75–84 may be due to motional coupling with the Arg/Ala-rich helix. A number of residues at the C-terminus also experience signal loss, suggesting additional membrane binding, possibly mediated by basic residues in this region, including K88, R96, and K99.

The solution NMR experiment is useful for identifying the helical region of FtsQ for membrane association, but the loss of crosspeaks precluded further characterization of FtsQ-membrane interactions. We thus turned to solid-state NMR spectroscopy using both magic-angle spinning (MAS) and oriented samples. When uniformly $^{13}C$-labeled FtsQ1-99 is bound to 7:3 POPG/POPC vesicles, residues that remain dynamic appear in an INEPT $^{13}C$-$^{13}C$ correlation spectrum acquired on MAS samples (Fig. 5a and Supplementary Fig. 5, blue crosspeaks). The $^{13}C$ chemical

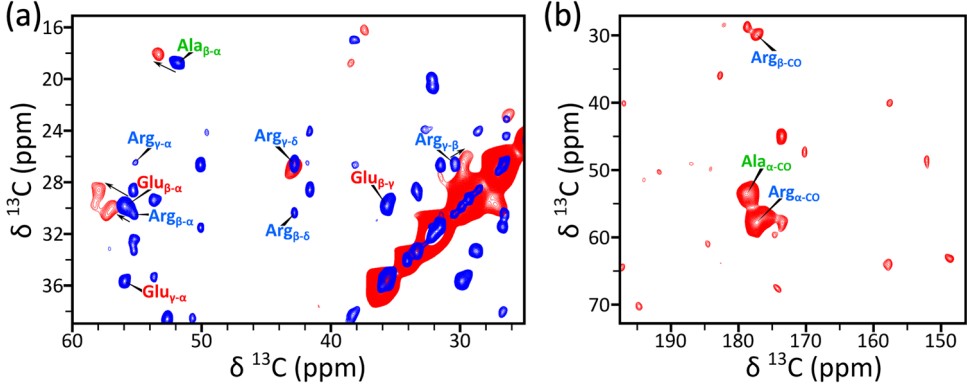

**Fig. 5 Identification of comparatively rigid and dynamic residues of FtsQ1-99 by MAS solid-state NMR. a** Overlay of INEPT (blue) and CP (red) $^{13}C$-$^{13}C$ correlation spectra. Acquired at 800 MHz with a 13 kHz spinning rate, employing 7.5 ms and 30 ms mixing times for TOBSY-INEPT (298 K) and CP-PARIS (285 K), respectively. The samples contained FtsQ1-99 bound to 7:3 POPG/POPC at a protein-to-lipid ratio of 1:20 in 20 mM Tris buffer (pH 6.85) with 50 mM NaCl. The selected region highlights the only significant CP-PARIS sidechain resonances, assigned to Ala, Arg, and Glu residues. **b** Carbonyl region of CP-PARIS highlighting significant Cα-CO cross peaks assigned to Ala and Arg residues. An Arg $C_\beta$-CO cross-peak is also present. Full spectra are shown in Supplementary Figs. 5 and 6.

shifts of these dynamic residues match well with those of nonhelical residues determined by solution NMR. For example, the $Ala_{\beta-\alpha}$, $Arg_{\beta-\alpha}$, and $Glu_{\beta-\alpha}$ cross peaks are at (51.83, 18.76), (55.34, 30.40), and (55.93, 29.86), respectively, and are close to the average $(C_\alpha, C_\beta)$ chemical shifts, $(52.3 \pm 0.1, 19.0 \pm 0.1)$, $(56.2 \pm 0.4, 30.6 \pm 0.1)$, and $(56.1 \pm 0.9, 30.0 \pm 0.3)$, of Ala, Arg, and Glu in nonhelical regions (Supplementary Table 2). In a cross-polarization (CP) $^{13}C$-$^{13}C$ correlation spectrum, which reports on residues that are rigid in membrane-bound FtsQ1-99, there are far fewer crosspeaks (Fig. 5a, red crosspeaks; Supplementary Fig. 6). The CP crosspeaks can be assigned to Ala, Arg, and Glu, with the help of spectra acquired on samples with only Ala and Arg residues labeled with $^{13}C$ (Supplementary Fig. 7). In particular, the $Ala_{\beta-\alpha}$, $Arg_{\beta-\alpha}$, and $Glu_{\beta-\alpha}$ CP crosspeaks move to (53.49, 18.10), (57.20, 30.10), and (58.00, 28.90) (Fig. 5a), matching well with the average $(C_\alpha, C_\beta)$ chemical shifts, $(54.0 \pm 0.2, 18.3 \pm 0.1)$, $(57.9 \pm 0.2, 30.1 \pm 0.2)$, and $(58.1 \pm 0.2, 29.4 \pm 0.2)$, of Ala, Arg, and Glu in the helical region (Supplementary Table 2). Crosspeaks of adjacent carbons farther along the Arg sidechain, i.e., $Arg_{\gamma-\beta}$ and $Arg_{\gamma-\delta}$, are also detected in the CP spectra (Fig. 5a and Supplementary Fig. 7). In the carbonyl carbon region (Fig. 5b and Supplementary Fig. 6), the CP $C^O$-$C_\alpha$ crosspeaks of Ala and Arg residues are very prominent, again at chemical shifts, (179.39, 53.22) and (177.71, 57.83), in good agreement with the counterparts of these amino acids in the helical region, $(179.7 \pm 0.3, 54.0 \pm 0.2)$, and $(177.8 \pm 0.4, 57.9 \pm 0.2)$. Lastly, we probed the sidechain nitrogens of rigid Arg residues by acquiring a 1-dimensional $^{15}N$ CP-MAS spectrum (Supplementary Fig. 8), which shows strong signals from $N_\varepsilon$ and $N_\eta$, in addition to that from the backbone nitrogens. In short, the INEPT and CP spectra reveal that, in membrane-bound FtsQ1-99, nonhelical residues are dynamic but residues in the Arg/Ala-rich helix are rigid, confirming that this helix is the major region participating in membrane binding.

We further characterized the membrane-bound FtsQ1-99 by solid-state NMR using uniformly $^{15}N$-labeled oriented samples. The PISEMA spectrum suggests a tilt angle of $74° \pm 4°$ for the Arg/Ala-rich helix with respect to the bilayer normal (Supplementary Fig. 9a). Moreover, the detection (by CP) of Arg sidechain $^{15}N$ anisotropic chemical shifts at 71 and 83 ppm (Supplementary Fig. 9b) again documents the significant role of Arg residues from the Arg/Ala-rich helix in membrane binding.

Closer inspection of the helical wheel projection of the Arg/Ala-rich helix (Fig. 1c) shows that the Ala ridge, formed by one Ala

residue from each of six successive turns (Ala53, Ala57, Ala60, Ala64, Ala68, Ala71), is confined to a 100° arc bordered by Arg49 and Arg50. The hydrophobic arc is widened somewhat by the $C_\beta$, $C_\gamma$, and $C_\delta$ methylenes of the bordering Arg residues (as well as by the $C_\beta$ and $C_\gamma$ methylenes of Gln67 projected into the same direction as Arg49). The single long nonpolar sidechain of Ile65 is not part of the hydrophobic arc. The result is an unusually narrow hydrophobic arc covered by only short nonpolar sidechains or portions thereof, suggesting a much shallower burial into the membrane hydrophobic core than typical amphipathic helices. This expectation is confirmed by MD simulations where the Arg/Ala-rich helix was initially buried to different depths into POPG/POPC membranes (Supplementary Fig. 10a). After 200 ns of simulations, the helix always came to the same shallowly buried depth (Supplementary Fig. 10b). In subsequent simulations lasting 1000 ns (16 replicates), the Arg/Ala-rich helix stably binds with the membrane (Fig. 6a).

Consistent with the PISEMA data, the helix has a tilt angle of $79° \pm 5°$ in the MD simulations, with the C-terminus more buried at the membrane interface than the N-terminus (Fig. 6a, b). Also consistent with the strong involvement of Arg residues from the Arg/Ala-rich helix in membrane binding revealed by both the MAS and orientated-sample solid-state NMR data, the Arg sidechains from the entire helix interact with lipid phosphates (Fig. 6c, d). With the N-terminus of the helix lifted, Arg49, Arg 50, and Arg52 reach downward for phosphates. Moving down the helix, Arg56 and Arg61 snorkel from the opposite sides of the helix, whereas Arg69 and Arg70 reach directly upward for phosphates. The sidechain-sidechain contact map of membrane-bound FtsQ1-99 shows that long-range contacts between the N-tail and the Arg/Ala-rich helix now all but disappear (Supplementary Fig. 11), because the Arg residues in the helix are already engaged with lipid phosphates. Local contacts between the last few residues of the N-tail and the first few residues of the Arg/Ala-rich helix are still formed, including salt bridges of Glu40 and Glu42 with Arg 55 and of Arg45 with Glu51, and may mediate motional coupling between the Arg/Ala-rich helix and flanking residues.

The solid-state NMR and MD simulation data presented above were obtained using model membranes, i.e., POPG/POPC membranes at a 7:3 molar ratio. To demonstrate that the results apply to the native membrane environment, we also carried out MAS and oriented-sample solid-state NMR studies of FtsQ1-99 binding to vesicles formed by native lipids of *M. smegmatis*, which is a nonpathogenic model of *Mtb*. The INEPT and CP

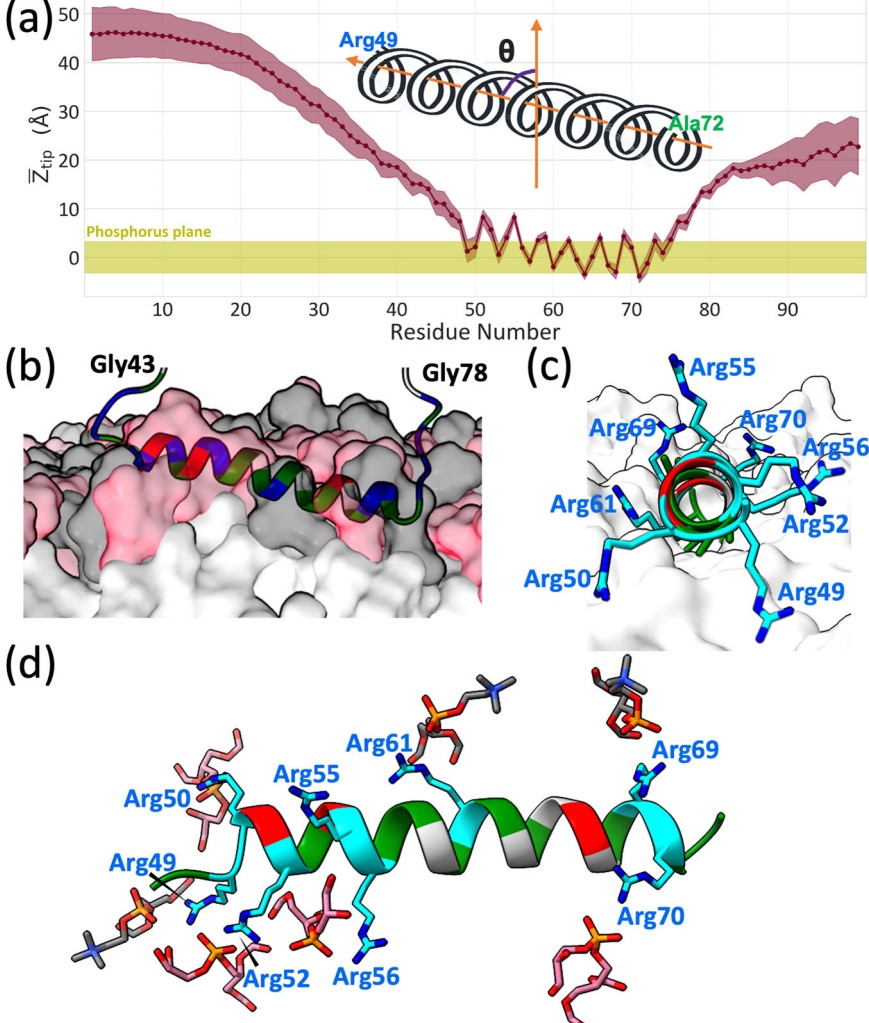

**Fig. 6 Interactions of FtsQ1-99 with acidic membranes revealed by MD simulations. a** Displacements of sidechain tip atoms from the average phosphorus plane in the upper leaflet. The mean $Z_{tip}$ values are shown as connected dots; standard deviations calculated over conformations collected from 16 replicate simulations are shown as a band. The root-mean-square-displacement of phosphorus atoms from the average plane is indicated by a band centered at $Z_{tip} = 0$. **b–d** Side, end, and top views of the Arg/Ala-rich helix bound to the membrane, with a tilt angle of 79°. Arg, Ala, Glu, and other residues are in blue or cyan, green, red, and gray colors, respectively. PG and PC headgroups and lipid acyl chains are displayed in **b** and **c** as surfaces in pink, gray, and white, respectively; PG and PC headgroups are displayed in **d** as sticks.

$^{13}$C-$^{13}$C correlation spectra obtained using model and native lipids superimpose very well (Supplementary Figs. 5, 6b), as do the PISEMA spectra (Supplementary Fig. 9c, d).

**The Arg/Ala-rich α-helix binds to the GTPase domain of FtsZ.** Because FtsZ and FtsQ are among the earliest proteins recruited to the divisome, we wondered whether the *Mtb* FtsQ N-terminal cytoplasmic region, with its much longer length than the *E. coli* counterpart, could bind to the water-soluble FtsZ. Our coelution assay shows that FtsQ1-99 indeed binds to *Mtb* FtsZ (Fig. 7a, b). More specifically, it binds to the GTPase domain (residues 1–312), not to the disordered C-tail (residues 313–379) of FtsZ. The participation of the FtsZ GTPase domain in binding with another divisome protein is unusual, as it is the C-tail that binds with FtsA, ZipA, and SepF for the membrane anchoring of FtsZ[11–13,16] and with FtsW at a later stage of cell division[20,21].

Solution NMR of FtsQ1-99 in the presence of *Mtb* FtsZ reveals the Arg/Ala-rich helix as the central region for binding to the FtsZ GTPase domain (Fig. 7c and Supplementary Fig. 12). Similar to the changes in peak intensities upon membrane binding

(Fig. 4b), resonances from the entire Arg/Ala-rich helix plus some flanking residues, from 38–79, are broadened beyond detection upon FtsZ binding. In addition, residues at the C-terminus also experience some loss of peak intensity, suggesting that they may form additional interactions with FtsZ. We confirmed that the FtsZ C-tail does not interact with FtsQ1-99 by solution NMR, as the HSQC spectrum of the former is unaffected by the presence of the latter (Supplementary Fig. 13). The solution NMR data thus shows that FtsQ1-99 binds to the GTPase domain of FtsZ, mainly through the Arg/Ala-rich helix.

We used computational modeling to identify the likely docking site for the FtsQ Arg/Ala-rich helix. This helix is very long and also highly positively charged. Both of these features are accommodated well by the cleft between the two subdomains of the FtsZ GTPase domain (Fig. 8a and Supplementary Fig. 14). At the base of this cleft is the core helix (named H7; residues 175–200[22]), which, with 26 residues, nearly matches the Arg/Ala-rich helix in length. Moreover, the electrostatic surface of the GTPase domain is mostly negative, and the negative electrostatic potential is the most intense in the inter-subdomain cleft. As a plausible pose, we docked the Arg/Ala-rich helix into this cleft in

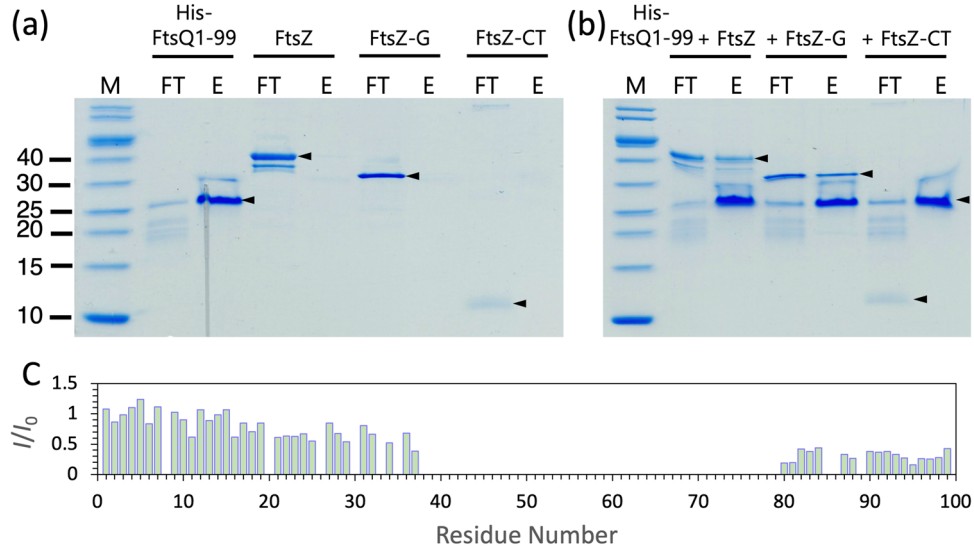

**Fig. 7 Coelution and solution NMR characterization of FtsQ1-99 binding with FtsZ. a** SDS-PAGE of aliquots collected from running His-tagged FtsQ1-99 (His-FtsQ1-99) and untagged full-length FtsZ or its GTPase domain (FtsZ-G) or C-tail (FtsZ-CT) individually through a nickel column. Abbreviations are: M, molecular-weight markers; FT, flow-through; E, elution. Arrowheads indicate the major population. **b** Corresponding results collected from running mixtures of His-FtsQ1-99 with FtsZ, FtsZ-G, or FtsZ-CT through a nickel column. Arrowheads indicate that FtsZ and FtsZ-G as well as His-FtsQ1-99 are found in the elution fractions but FtsZ-CT is only found in the flow-through fraction. **c** Intensity ratio of crosspeaks in the $^{1}$H-$^{15}$N HSQC spectra of 50 μM uniformly $^{13}$C–$^{15}$N labeled FtsQ1-99 in the presence (*I*) and absence (*I_0*) of 50 μM FtsZ. Acquired at 800 MHz and 298 K in 20 mM phosphate buffer (pH 6.5) with 25 mM NaCl.

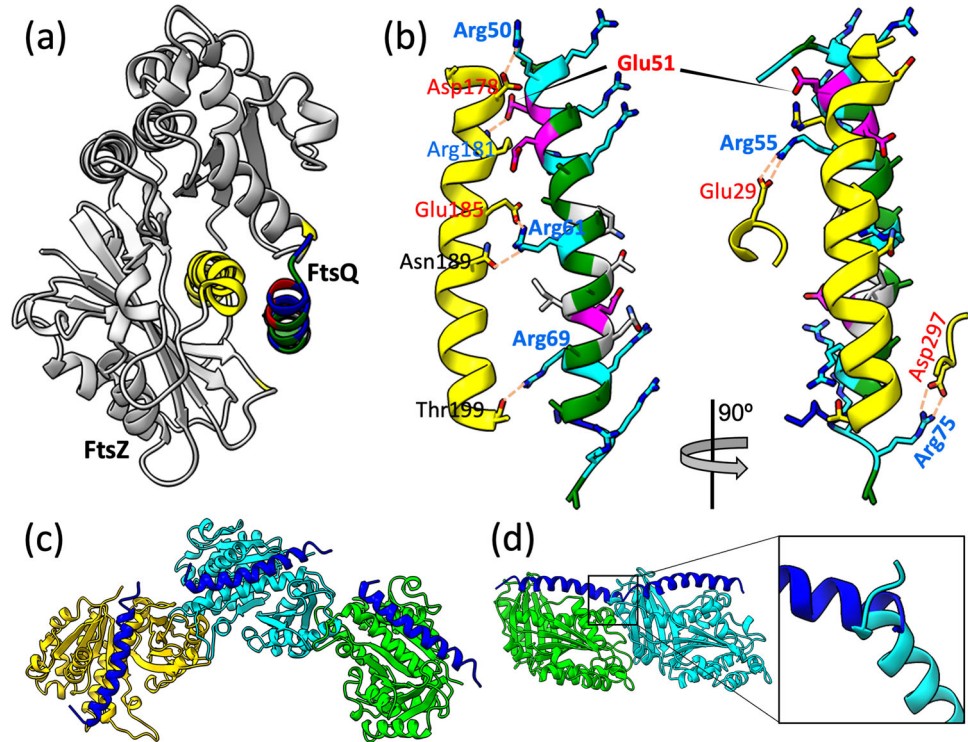

**Fig. 8 Modeled complex between the FtsQ Arg/Ala-rich helix and monomeric or polymeric FtsZ. a** The FtsQ helix docked to the GTPase domain of FtsZ (Protein Data Bank (PDB) entry 1RQ2[22]). The FtsQ helix is shown with Arg, Ala, Glu, and other residues in blue, green, red, and gray, respectively. The FtsZ GTPase domain is shown in gray, except with yellow for the H7 helix at the base of the inter-subdomain cleft and two short segments at the entrance of the cleft. **b** Salt bridges and hydrogen bonds between the FtsQ helix and the inter-subdomain cleft of the FtsZ GTPase domain. **c** The FtsQ helix docked to a GTPase domain in a curved FtsZ polymer (PDB entry 4KWE[26]), free of clashes with neighboring GTPase domains. **d** The FtsQ helix docked to a GTPase domain in a straight FtsZ polymer (PDB entry 1W5A[64]), clashing with neighboring GTPase domains (see inset).

a parallel direction with respect to the H7 helix and with an array of Arg residues, at positions 50, 61, and 69, within the interface (Fig. 8b, left panel). A number of salt bridges and hydrogen bonds are formed between the two helices, including FtsQ Arg50 with

FtsZ Asp178, FtsQ Glu51 with FtsZ Arg181, FtsQ Arg61 with FtsZ Glu185 and Asn189, and FtsQ Arg69 with FtsZ Thr199. The Arg/Ala-rich helix also forms two additional salt bridges with the walls of the cleft, i.e., between FtsQ Arg55 and FtsZ Glu29 on one

side and between FtsQ Arg75 and FtsZ Asp297 on the opposite side (Fig. 8b, right panel). Regardless of the precise docking site, direct binding of the FtsQ Arg/Ala-rich helix to the FtsZ GTPase domain keeps the Z-ring in close proximity to the inner membrane.

## Discussion

By combining solution and solid-state NMR, MD simulations, and other techniques, we have thoroughly characterized the conformational and dynamic properties of the 99-residue N-terminal cytoplasmic region of FtsQ in solution and its binding to acidic membranes and to FtsZ. The most notable feature is a 29-residue Arg/Ala-rich α-helix that can interact with three partners: N-terminal residues in the same domain, acidic membranes, and the GTPase domain of FtsZ. The latter property makes FtsQ a potential membrane anchor or at least a stabilizer of the *Mtb* Z-ring.

While FtsQ is also expressed in *E. coli*, its N-terminal cytoplasmic region is only about one-quarter the length of the *Mtb* homolog, basically enough to maintain the N-terminus on the cytoplasmic side of the membrane and unable to bind FtsZ. Instead, membrane anchoring in *E. coli* is achieved by two other proteins: FtsA and ZipA. *Mtb* does not have homologs of these membrane anchors. It now seems possible that their role can be partly fulfilled by FtsQ. A retracted paper claimed that FhaB acted as a membrane anchor of the Z-ring by bridging between FtsQ and FtsZ[23], but here we show that FtsQ and FtsZ directly bind to each other. Initially, FtsW was thought to be a membrane anchor of the Z-ring[20], but subsequently, the FtsZ-FtsW interaction was found to regulate the biogenesis of cell wall[21]. Another protein, SepF, has already been characterized as a membrane anchor of the Z-ring in *B. subtilis*, *C. glutamicum*, and *Mtb*[11,14–17]. SepF is a water-soluble protein but tethers to membranes by an amphipathic helix at its N-terminus and, in the *Mtb* version, also Arg-rich stretches in the disordered linker preceding the FtsZ-binding core domain[17]. The additional membrane interactions by the *Mtb* SepF linker bring the Z-ring into close proximity to the membrane. Therefore, both FtsQ and SepF, either together or independently, keep the *Mtb* Z-ring close to the membrane. The membrane proximity in turn allows the Z-ring to interact with and thus recruit other membrane proteins including FtsW[20,21] and CrgA[24] at a later stage of cell division. FtsQ also interacts with CrgA[24], possibly through their transmembrane helices; the tripartite interactions between FtsQ, CrgA, and the Z-ring help maintain the integrity of the late divisome.

In contrast to other divisome proteins that bind to FtsZ, FtsQ binds to the GTPase domain instead of the disordered C-tail of FtsZ. FtsQ binding to the GTPase domain potentially impacts drug binding to the same domain as well as other functions of this domain, and therefore has important consequences. In particular, the C-terminal half of the H7 helix in the GTPase domain lines a drug-binding site[25]. If the FtsQ Arg/Ala-rich helix indeed binds to the inter-subdomain cleft as modeled here (Fig. 8a), then drug binding and FtsQ binding would interfere with each other. This scenario would have implications for drug development against tuberculosis. We cannot exclude the possibility that FtsQ also binds to the C-tail of FtsZ when it is present at high concentrations in the Z-ring. *E. coli* ZipA was recently suggested to interact with both the C-tail and the GTPase domain[13].

Another important issue raised by the fact that FtsQ binds to the GTPase domain concerns the formation of the Z-ring. Because the Z-ring is formed by the polymerization of the GTPase domain, FtsQ binding could affect the polymerization process. It has been recognized that, upon initial GTP binding, FtsZ polymerizes in a straight manner, but after GTP hydrolysis

into GDP, the FtsZ polymers become curved, thereby facilitating Z-ring formation[26]. Interestingly, in the binding pose modeled here (Fig. 8a), the FtsQ Arg/Ala-rich helix does not interfere with the GTPase domains of the adjacent FtsZ monomers in curved polymers (Fig. 8c) but encounters clashes in straight polymers (Fig. 8d). In that case, FtsQ would not only anchor the Z-ring to the membrane but also stabilize its curvature.

## Materials and methods

**Protein expression and purification.** *Mtb* FtsQ1-99 with an N-terminal His-tag plus a TEV cleavage site was expressed in *E. coli* BL21 cells. For expression of $^{13}$C-$^{15}$N labeled protein, cells were grown in LB media at 310 K until OD at 600 nm reached 0.7. Cells were then transferred to M9 media supplemented with 1 g of $^{15}$N ammonium chloride and 2 g $^{13}$C D-glucose per liter of culture for isotope labeling. Cultures were incubated for 2 h at 298 K before adding IPTG at a 0.4 mM final concentration. Protein expression continued overnight at 298 K. Expression of unlabeled protein proceeded in LB media. Cells were resuspended in 20 mM Tris (pH 8.0) and 500 mM NaCl and then lysed using French press. Cellular debris was removed by ultracentrifugation at $100,000 \times g$ at 281 K for 1–2 h. Clarified lysate was loaded into a column prepared with Ni-NTA resin (Qiagen). Resin was washed with lysis buffer containing 60 mm imidazole and then protein was eluted with 400 mM imidazole. Fractions containing FtsQ1-99 were collected and mixed with 1 mg of TEV protease. This sample was dialyzed overnight at 277 K against a 20 mM Tris (pH 8.0) buffer with 500 mM NaCl. To remove TEV protease, the dialyzed sample was loaded to a Ni-NTA column where FtsQ1-99 was collected from the flow-through and washes with 20 mM imidazole. Fractions with FtsQ1-99 were dialyzed against either a 20 mM Tris buffer (pH 6.85) with 50 mM NaCl or a 20 mM sodium phosphate buffer (pH 6.5) with 25 mM NaCl. Samples that required additional purification were subjected to anion exchange using a Q Sepharose column. FtsQ1-99 samples were affixed and washed with either 20 mM Tris (pH 7.4) or 20 mM sodium phosphate buffer (pH 6.5) and subsequently eluted under a NaCl gradient.

Similarly, *Mtb* FtsZ and its globular domain (residues 1–312; FtsZ-G) and C-tail (residues 313–379; FtsZ-CT) were expressed with an N-terminal His-tag plus a TEV cleavage site. Expression and purification of FtsZ constructs proceeded in a similar fashion as FtsQ1-99, except for a different procedure during anion exchange chromatography. A wash with 300 mM NaCl and elution with 500 mM NaCl were used for full-length FtsZ and FtsZ-G, and no anion exchange was used for FtsZ-CT. Protein concentrations were estimated using Bradford assay.

**Solution NMR spectroscopy.** All solution NMR experiments were performed at 800 MHz using a TCI ($^1$H/$^{13}$C/$^{15}$N) cryoprobe equipped to an Avance II console operated with TopSpin 2.1. Sequential backbone assignment for FtsQ1-99 was carried out using 3D HNCO, HN(CA)CO, HNCACB, CACB(CO)NH experiments at 305 K in 20 mM Tris (pH 6.85) with 50 mM NaCl and 7.5% D$_2$O. Data were processed and analyzed using NMRPipe[27] and Sparky[28].

$^1$H-$^{15}$N J-couplings and RDCs were collected using an IPAP-WATERGATE HSQC sequence[29] with partial alignment achieved using 6 mm stretched acrylamide gels[30]. Samples contained 240 μM of uniformly $^{15}$N-labeled FtsQ1-99 in 20 mM Tris buffer (pH 6.85) with 50 mM NaCl. In-phase and anti-phase spectra were collected at 800 MHz and 305 K under both isotropic and partially aligned conditions. RDC values were extracted using NMRPipe and Sparky.

**Paramagnetic relaxation enhancement experiments.** Three FtsQ1-99 cysteine mutants were prepared for MTSL spin labeling. Mutations were localized in the N-tail (T24C), the Arg/Ala-rich α-helix (T63C), and the C-tail (A87C). Each uniformly $^{15}$N-labeled mutant was purified as described above but in the presence of 1 mM DTT. FtsQ1-99 mutants were incubated for 4 h with ×20 excess of MTSL spin label (Santa Cruz Biotechnology) in 20 mM phosphate buffer (pH 6.5) with 25 mM NaCl and 0.5 mM DTT. Additional MTSL was added to the samples and incubated overnight at room temperature. Samples were dialyzed against a 20 mM phosphate buffer (pH 6.5) with 25 mM NaCl to remove unbound MTSL. 2D $^1$H-$^{15}$N HSQC spectra were collected for each 50 μM MTSL-labeled sample. After these experiments, diamagnetic samples were prepared by adding 2 mM sodium ascorbate (Sigma-Aldrich) and incubating for 1 h at room temperature. 2D $^1$H-$^{15}$N HSQC of the reduced samples were collected in the same conditions as before. PRE effects were estimated by the signal intensity ratio between the paramagnetic and diamagnetic samples ($I_{para}/I_{dia}$).

**Changes in NMR peak intensities upon binding.** Binding of FtsQ1-99 with lipids was assessed by collecting 2D $^1$H-$^{15}$N HSQC spectra of 130 μM uniformly $^{13}$C-$^{15}$N-labeled FtsQ1-99 in the absence and presence of POPG/POPC (7:3 molar ratio) vesicles at protein-to-lipid ratios of 1:5 (0.65 mM lipids) and 1:10 (1.30 mM lipids). Samples were prepared in 20 mM Tris (pH 6.85) with 50 mM NaCl and 10% D$_2$O. Data were collected at 800 MHz and 305 K. Similarly, binding to FtsZ was assessed by adding 50 μM of unlabeled FtsZ to the same amount of uniformly $^{13}$C-$^{15}$N labeled FtsQ1-99. Binding was monitored by the changes in peak intensity.

In addition, possible interactions of FtsQ1-99 with FtsZ C-tail were evaluated by collecting 2D $^1$H-$^{15}$N HSQC spectra of 50 µM uniformly $^{13}$C-$^{15}$N labeled FtsZ C-tail in the absence and presence of 75 µM FtsQ1-99. Similar conditions were used as described above.

**Backbone amide relaxation experiments**. Backbone $^{15}$N $R_1$ rates were measured with 7 delay points between 50 ms and 1.0 s. Cross-correlated relaxation was suppressed with a WALTZ decoupling scheme during $^{15}$N relaxation using a $^1$H field of 5000 Hz. $^{15}$N $R_{1\rho}$ rates were measured with a 1500 Hz spin-lock field using delays of 8 ms to 200 ms. Cross-correlated relaxation was suppressed using $^1$H 180° decoupling pulses with a 22.1 kHz field strength every 2 ms during the spin-lock period. The $R_1$ and $R_{1\rho}$ rates were calculated by fitting the intensity decay profiles to a mono-exponential using Sparky[28]. $R_2$ rates were calculated from the measured $R_{1\rho}$ rates using the effective field and spin-lock tilt for each $^{15}$N resonance, and correction for the contribution from the $R_1$ rate[31]. A 4-second relaxation delay was used between experiments. $^{15}$N[$^1$H] NOEs were measured using the pulse sequence of Farrow et al.[32], with a relaxation delay of 8 s between experiments.

**Solid-state NMR spectroscopy**. $^{15}$N-detected magic angle spinning (MAS) spectra were collected at 600 MHz using a custom-built low-E 3.2 mm triple channel probe equipped to an Avance Neo console operated with the Topspin 4.0 software. All samples were spun at 8 kHz using a Bruker MAS III control unit. $^{15}$N chemical shift referencing was accomplished under identical conditions using an NH$_4$Cl powder sample calibrated to 39.3 ppm. One-dimensional $^{15}$N-detected spectra were measured at calibrated a temperature of 295 K with a $^1$H-$^{15}$N contact time of 1 ms and $^1$H-decoupling achieved using a 75 kHz RF field applied with a SPINAL64 scheme.

Two-dimensional MAS measurements were performed at 800 MHz using a custom-built low-E 3.2 mm triple channel probe equipped to an Avance III-HD console operated with the Topspin 3.2 software. All samples were spun at 13 kHz using a Bruker MAS II control unit. $^{13}$C chemical shift referencing was accomplished under identical conditions using the carbonyl carbon of a crystalline glycine sample set to 178.4 ppm. $^{13}$C-$^{13}$C phase-altered recoupling irradiation scheme (PARIS) spectra[33] were measured at a calibrated temperature of 285 K with a $^1$H-$^{13}$C contact time of 700 µs and a $^{13}$C-$^{13}$C mixing period of 30 ms. $^1$H-decoupling was accomplished with a 100 kHz RF field applied with a SPINAL64 scheme. $^{13}$C-$^{13}$C total through-bond-correlation spectroscopy (TOBSY) spectra[34,35] were collected at a calibrated temperature of 298 K with a 7.5 ms mixing period.

To prepare mechanically oriented samples, POPG and POPC powders were weighed out in 7:3 molar ratio and dissolved in 2:1 chloroform/methanol. This solution was dried under a stream of nitrogen in a glass round-bottom flask. Residual organic solvent was removed by overnight lyophilization. Lipid films were rehydrated in 20 mM Tris (pH 6.85) and 50 mM NaCl and bath sonicated to produce small unilamellar vesicles. FtsQ1-99 suspended in matching buffer was added to the vesicle solution and incubated at room temperature for 1-2 h. Proteoliposomes were spread over ~40 glass slides and allowed to dry in a sealed box. A droplet of 1 µl water was applied to each slide before being stacked and subsequently stored in 97% relative humidity overnight. The stack of glass slides was than inserted into a rectangular glass tube, and sealed with wax and parafilm.

Oriented-sample solid-state NMR measurements were performed at 800 MHz using a custom-built low-E static $^1$H-X-Y flat coil probe and an Avance I console operated with Topspin 2.1. PISEMA spectra were collected at 298 K using a "Magic Sandwich" pulse sequence[36]. The RF fields for both $^1$H and $^{15}$N channels during cross-polarization (CP) and SAMMY blocks were 50 kHz with a CP contact time of 0.7 ms. $^1$H decoupling was applied at 30 kHz during direct dimension evolution with a recycle delay of 4 s. $^{15}$N chemical shifts were referenced to a gramicidin standard and the dipolar scaling factors were taken from Cui et al.[37].

**Native lipid purification**. Native lipids from *M. smegmatis* were purified using a sucrose density gradient and Folch lipid purification[38,39]. In short, mycobacterial cells were lysed first by French press followed by high-pressure homogenization with an Emulsiflex C5 system. Lysate was first centrifuged at $10,000 \times g$ to remove cell wall components, then at $27,000 \times g$ to remove large cellular debris, and finally at $100,000 \times g$ to condense the plasma membrane. The membrane fraction was then purified using a sucrose density gradient at $100,000 \times g$ whereafter lipids were further purified from residual protein components using a Folch method. Lipids separated in this way were suspended in a 2:1 (v/v) chloroform-methanol mixture and subsequently dried under a stream of N$_2$ gas in a glass round-bottom flask to form lipid films.

**Modeling of FtsQ1-99 solution structure**. 100 models were generated using XPLOR-NIH version 3.0[40] and imposing RDC and TALOS+[41] predicted dihedral restraints. Each simulated annealing trajectory followed a linear temperature ramp from 1000 K to 10 K in 10 K steps. Force constants for dihedral-angle restraints were held constant at 300 kcal mol$^{-1}$ rad$^{-2}$. Force constants for RDC restraints were ramped from 0.001 kcal mol$^{-1}$ Hz$^{-2}$ to 0.08 kcal mol$^{-1}$ Hz$^{-2}$. Eight lowest-energy models were saved.

**Small-angle X-ray scattering**. SAXS data for FtsQ1-99 was collected at the Advanced Photon Source of the Argonne National Laboratory using the DND-CAT 5ID-D beam line. Data were collected on two FtsQ1-99 samples, one at 0.75 mg/mL and the other at 1.5 mg/mL. These samples were prepared in 20 mM phosphate buffer (pH 6.5) with 25 mM NaCl. Data from these two samples were merged using ATSAS software package.

**Pulldown and coelution assays**. Lipid binding was assayed by mixing 320 µM FtsQ1-99 with lipid vesicles (6.58 mM lipids; protein-lipid ratio ~1:20) in 20 mM Tris (pH 6.85) with 50 mM NaCl. After ~2 h of incubation, mixtures were subjected to centrifugation at $100,000 \times$ rpm for 4–12 h at 281 K to condense vesicles. Immediately thereafter, supernatant aliquots were collected directly from the centrifuge rotor to avoid condensate redispersion. Samples were then visualized using SDS-PAGE electrophoresis.

FtsZ binding was assayed by mixing 50 µg of His-tagged FtsQ1-99 with 50 µg of FtsZ, FtsZ-G, or FtsZ-CT in 20 mM phosphate buffer (pH 6.5) with 25 mM NaCl in a final volume of 500 µL. These samples were loaded onto a Ni-NTA column (100 µL) and the flow-through was collected. After column washing with buffer including 20 mM imidazole, His-FtsQ1-99 was eluted with buffer containing 400 mM imidazole. Each protein was tested by itself as a negative control. Aliquots of flow-through and elution were evaluated by SDS-PAGE.

**Molecular dynamics simulations**. MD simulations of FtsQ1-99 in solution were similar to previous studies of disordered proteins[18,42], using the AMBERff14SB force field[43] for proteins and the TIP4PD water model[44] The starting structure for the simulations was the lowest-energy model calculated by XPLOR-NIH as described above. Once solvated, Na$^+$ and Cl$^-$ were added to neutralize the system and achieve a 50 mM salt concentration, resulting in system dimensions of 126 Å × 126 Å × 179 Å and a total of 358,572 atoms. The system was energy minimized (2000 steps of steepest descent and 3000 steps of conjugate gradient) and heated to 300 K at constant volume over 100 ps at 1 fs timestep in sander. The simulations then continued on GPUs in 12 replicates using pmemd.cuda[45] at constant temperature (300 K) and pressure (1 atm) for 1134 ns at 2 fs timestep. Temperature was regulated by the Langevin thermostat[46] with a 3.0 ps$^{-1}$ damping constant; pressure was regulated by the Berendsen barostat[47] with a relaxation time of 0.5 ps. Bond lengths involving hydrogen atoms were constrained using SHAKE[48]. Long-range electrostatic interactions were treated by the particle mesh Ewald method[49], with a nonbonded cutoff distance of 10 Å. The final 1000 ns, with snapshots saved at 20 ps intervals, was used for analysis.

MD simulations of FtsQ1-99 bound to membranes were similar to a previous study[50]; details identical to those in solution are not repeated. The forcefield for lipids was Lipid17[51]. The initial structure of FtsQ1-99 was obtained from the simulations in solution, with the Arg/Ala-rich helix embedded into membranes at three different depths using the CHARMM-GUI webserver[52]. The membranes were composed of 210 POPG lipids and 90 POPC lipids per leaflet. The system dimensions were 146 Å × 146 Å × 159 Å, with a total of 426,384 to 446,918 atoms. Each system was energy-minimized (10,000 steps of conjugate gradient) and equilibrated in NAMD 2.12[53] following a CHARMM-GUI six-step protocol[52]. This protocol consisted of three steps of 25 ps each at 1 fs timestep and three steps of 100 ps each at 2 fs timestep. The first two steps were at constant temperature and volume whereas the remaining four steps were at constant temperature and pressure; restraints on the protein and lipids were gradually released in the six steps. Temperature was maintained at 300 K by the Langevin thermostat[46] with a damping constant of 1.0 ps$^{-1}$ and pressure was kept at 1 atm by the Langevin piston[54] with an oscillation time period of 50 fs and decay time of 25 fs. The nonbonded cutoff distance was 12 Å with force switching at 10 Å. The simulations of each system were continued on GPUs in four replicates for 200 ns at constant temperature and pressure. The nonbonded cutoff distance was 11 Å with force switching at 9 Å. Semi-isotropic scaling was applied in the x-y direction to maintain the overall shape of the system. At the end of these simulations, the systems converged to the same burial depth (Supplementary Fig. 10b). Sixteen replicate simulations continued for another 1000 ns. These simulations data were used previously to train and test a predictor for membrane-association propensities[55].

**Data analyses for MD simulations**. Data analyses followed previous work[18,42,50]. Cpptraj[56] from Ambertools19 was used to calculate secondary structures (implementing the DSSP algorithm[57]). Secondary structures were further modified to incorporate the formation of polyproline II (PPII) stretches when three or more consecutive residues were classified as coil and fell into the PPII region of the Ramachandran map. To calculate contact maps, MD trajectory files were loaded using mdtraj[58] and distances between the sidechain heavy atoms (except those from the same residue) were calculated. A contact was defined if two heavy atoms were within 3.5 Å. The number of snapshots in which a contact was formed, divided by the total number of snapshots, gave the contact frequency. To calculate NMR relaxation properties, the time correlation function of each backbone NH bond vector was first fit to a sum of three exponentials, and the resulting Fourier transform was used in standard formulas to calculate $R_2$, $R_1$, and NOE. To compensate for the effect of the Langevin thermostat on dynamic properties,

the time constants from the three-exponential fit were scaled by pre-determined factors[59].

$Z_{tip}$ distances were also calculated using cpptraj. For each residue, $Z_{tip}$ was defined as between the carbon atom farthest along the side chain and the average phosphorus plane of the upper leaflet. To define the tilt angle of the helix core, the centers of geometry of three $C_\alpha$ atoms (residues 49–51) in the first turn and the last turn (residues 70–72) were calculated; the angle between the line connecting these two points and the normal vector of the average phosphorus plane was the tilt angle.

The following properties were calculated using every 10th of the saved snapshots. SHIFTX2[60] was used to calculate the chemical shifts. Random-coil chemical shifts used for calculating secondary chemical shifts were from POTENCI[61]. DEER-PREdict[62] was used to calculate PREs. FoXS[63] was used to calculate the SAXS profile, with the calculated result scaled for each snapshot to minimize deviation from the experimental data.

**Statistics and reproducibility**. Experimental data (e.g., NMR, CD, and pulldown) were acquired according to common practices. Molecular dynamics simulations were carried out for 12 replicates in solution and for 16 replicates at the membrane surface. Convergence of the MD simulations was validated by showing that the results calculated in 250-ns blocks along the trajectories fall within 1 standard deviation of the mean. The mean and standard deviation were calculated using the replicate simulations.

**Reporting summary**. Further information on research design is available in the Nature Portfolio Reporting Summary linked to this article.

## Data availability

All data generated or analyzed during this study are included in this published article (and its supplementary data files). The chemical shifts and relaxation data of FtsQ1-99 in solution have been deposited to Biomolecular Magnetic Resonance Database (https://bmrb.io/) under accession code 51860. The SAXS data have been deposited to the Small Angle Scattering Biological Data Bank (https://www.sasbdb.org/) under the accession code SASDRA6. The source data for all plots presented in the figures are uploaded as Supplementary Data 1.

## Code availability

Data analysis procedures were described under Materials and Methods. All the computer programs used were cited and publicly available. The input files for MD simulations in solution and at the membrane surface and the initial and final coordinate files are uploaded as Supplementary Data 2–7.

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

## Acknowledgements
This work was supported by National Institutes of Health Grants AI119187, GM122698, and GM118091. All NMR experiments were carried out at the National High Magnetic Field Laboratory supported by the NSF Cooperative Agreement No. 1644779 and the State of Florida.

## Author contributions
T.A.C. and H.-X.Z. designed the research; S.T.S. and C.A.E., S.D. performed the research and analyzed the data; T.A.C. and H.-X.Z. wrote the manuscript.

## Competing interests
The authors declare no competing interests.
