## [Peer Review File · Communications Biology]

Reviewers' comments:

Reviewer #1 (Remarks to the Author):

Reviewer comments for

"An Arg/Ala-Rich Helix in the N-Terminal Region of *M. tuberculosis* FtsQ Anchors FtsZ to Membranes"

In this study, Smrt et. al have studied the mainly disordered cytoplasmic 99- residue N-terminus of ftsQ protein and identified a 29-residue α -helical region. They show this Arg/Ala-rich helix interacts with the GTPase domain of ftsZ protein and acts as a membrane anchor for the Z-ring formation relevant to Mtb cell division. The authors have used a combination of structural biology techniques including solution NMR, solid-state NMR, molecular dynamics, etc. to study the structure of the 29-residue region in different conditions. Using solution NMR, the authors have assigned the backbone residues and come up with the helical wheel suggesting an Arg/Ala-rich 29-residue fragment as an interpretation of this data, based on r1 and r2 relaxation and NOEs of isotopically labeled protein. Using MD simulations, they have corroborated this theory, and also tested the binding of this region to an acidic inner membrane. Using PRE experiments by site-specific spin labeling, the authors have studied the intramolecular interactions within the same protein domain. To test the membrane binding experimentally, the authors have used POPG: POPC 7:3 as a standard membrane-mimicking environment. They have used ssNMR to corroborate these interactions. Based on their findings, the authors propose that FtsQ and GTPase domain of FtsZ directly interact, and the Arg/Ala-rich helical domain acts as a membrane anchor for the formation of the Z-ring. This work is a comprehensive study to demonstrate α -helical structure within FtsQ protein in an otherwise disordered domain and proposes its role in intramolecular interactions with upstream amino acid residues and intermolecular interactions with membrane lipids.

Comments:

- The caption for Figure 1 states that the 15N HSQC was acquired with 25 mM NaCl, while as per the methods section, 3D experiments were done in 50 mM NaCl. How do the authors explain their reasoning behind this change in salt concentration for 2D and 3D experiments of uniformly double-labelled protein?
 - Why was 305 K selected temperature for solution NMR to acquire the 3D experiments? Have the authors tested how the spectra (possibly a quick HSQC) differ at room temperature or physiological temperature of 37 C? Do their findings vary?
 - The authors should include 1-2 sentences explaining the gaps in experimental RDCs obtained for readers who do not have a background in NMR to convey the limitations of the experiments. For e.g., residues near 40-50 sites in different sections of Fig 2. It is relevant since the beginning ~48 of the proposed helix lies within this region.
 - Since the authors refer to the values from Abyzov et al (ref. 18) in support of their helix in a disordered protein model, could they comment on the temperature-dependency of this structure? Similar to the R1, R2, and NOE values obtained in ref. 18, the authors could use the RDCs and time scale information of motion to comment on the flexibility or rigidity of their proposed helix and how it may affect its interactions.
 - What was the rationale for the authors to shortlist T24, T63, and A87 as specific locations for spin labeling? How do they think changing these sites, for e.g. T29, may impact their findings?
 - For ssNMR experiments, why were the calibrated temperature kept different during the experiments, what was the estimated sample temperature in all these cases?
- What is the stoichiometric ratio of FtsQ and FtsZ proteins identified through co-elution? How does it correlate to the physiological ratios of these proteins in Mtb divisome?
- How do the authors interpret 'motional coupling' to explain the loss of peak intensity for residues 21-45 and 75-84 with the Arg/Ala-rich helix in presence of POPG/POPC vesicles?
 - In the discussion, the authors mention the potential of their findings in drug binding. It would be nice to test a standard drug and tested how their interactions were affected. Are there any similar

studies available in the literature? Without any experiments, since it is possible that the drugs do not interact at all, or bind through an entirely different mechanism, currently the discussion seemingly oversells this point.

- Since the length of the helix is almost thrice a generic helical length, which conditions support stabilizing this specific conformation in the otherwise disordered protein?
- Although a proposed structure is simulated computationally based on the findings, the authors could include a comment on what may be missing from the experimental data to build a structure for this proposed helix and outline the challenges of experimentally converting this proposed helical wheel into a well-resolved structure. This emphasizes the need for such nicely done comprehensive studies.

Minor corrections:

- Figure 1 A: Helical region is underlined from 46-74, whereas the helical wheel included 48-72 residues. Why is a discrepancy between experimental and MD data?
- Figure 1 B: Peaks assigned to H38 and N85 are not visible in the figure, authors may check the arrows are marked correctly. Is the peak next to A48 not assigned to any residue?
- The last word in the main text referencing Table S 2, should be 'non-helical' instead of nonlocal.
- Table S 2: What does superscript 2 refer to (next to nonhelical) ?
- Fig. S 4: The labels of I10 and A72 are interchanged. In the caption, POPC is misspelled.
- I suggest adding an introductory sentence to the results of ssNMR experiments in this study. At present, it seems as though a sentence is missing before this paragraph in the results.

Reviewer #2 (Remarks to the Author):

In this work, the authors used NMR and MD simulation to study the conformation and binding behavior of Arg/Ala-rich region of Mtb FtsQ to its N-terminal region, acidic membrane and GTPase domain of FtsZ. The Arg/Ala-rich helix is adopted, it can effectively bind to GTP domain of FtZ and affect the formation of Z-ring. Hence, The FtsQ may serve as a membrane anchor of Z-ring. The study about the folding-upon-binding behavior of IDR and the response to various binding partner is a very interesting topic and has apparent biomedical relevance, for example the investigation about the additional binding partner of IDR. With this regards, the binding behavior and conformation of Arg/Ala-rich helix with different partner should be compared in a more quantitative way, e.g. binding stability, interaction mode and the critical residues for protein interaction. Moreover, the discussion about the mechanism underlying the folding-upon-binding behaviors of this IDR under various environment is recommended.

We thank the reviewers for the thoughtful comments. Below we present our point-by-point response in blue.

Referee expertise:

Referee #1: NMR, IDPs

Referee #2: MD simulations

Reviewers' comments:

Reviewer #1 (Remarks to the Author):

Reviewer comments for

"An Arg/Ala-Rich Helix in the N-Terminal Region of *M. tuberculosis* FtsQ Anchors FtsZ to Membranes"

In this study, Smrt et. al have studied the mainly disordered cytoplasmic 99- residue N-terminus of ftsQ protein and identified a 29-residue α -helical region. They show this Arg/Ala-rich helix interacts with the GTPase domain of ftsZ protein and acts as a membrane anchor for the Z-ring formation relevant to Mtb cell division. The authors have used a combination of structural biology techniques including solution NMR, solid-state NMR, molecular dynamics, etc. to study the structure of the 29-residue region in different conditions. Using solution NMR, the authors have assigned the backbone residues and come up with the helical wheel suggesting an Arg/Ala-rich 29-residue fragment as an interpretation of this data, based on r1 and r2 relaxation and NOEs of isotopically labeled protein. Using MD simulations, they have corroborated this theory, and also tested the binding of this region to an acidic inner membrane. Using PRE experiments by site-specific spin labeling, the authors have studied the intramolecular interactions within the same protein domain. To test the membrane binding experimentally, the authors have used POPG: POPE 7:3 as a standard membrane-mimicking environment. They have used ssNMR to corroborate these interactions. Based on their findings, the authors propose that FtsQ and GTPase domain of FtsZ directly interact, and the Arg/Ala-rich helical domain acts as a membrane anchor for the formation of the Z-ring. This work is a comprehensive study to demonstrate α -helical structure within FtsQ protein in an otherwise disordered domain and proposes its role in intramolecular interactions with upstream amino acid residues and intermolecular interactions with membrane lipids.

Comments:

- The caption for Figure 1 states that the ^{15}N HSQC was acquired with 25 mM NaCl, while as per the methods section, 3D experiments were done in 50 mM NaCl. How do the authors explain their reasoning behind this change in salt concentration for 2D and 3D experiments of uniformly double-labelled protein?

The Fig. 1 capture made a typo – it should be 50 mM NaCl in both experiment, and we have now made the correction.

- Why was 305 K selected temperature for solution NMR to acquire the 3D experiments? Have the authors tested how the spectra (possibly a quick HSQC) differ at room temperature or physiological temperature of 37 C? Do their findings vary?

The choice of 305 K is common when investigating bacterial proteins because this is a temperature at which they are cultured and typically have the highest growth rate. It is slightly lower, yet very close to human body temperature, where the pathogens are also active.

- The authors should include 1-2 sentences explaining the gaps in experimental RDCs obtained for readers who do not have a background in NMR to convey the limitations of the experiments. For e.g., residues near 40-50 sites in different sections of Fig 2. It is relevant since the beginning ~48 of the proposed helix lies within this region.

We have added 2 sentences to explain the limitations of NMR experiments (p. 5).

- Since the authors refer to the values from Abyzov et al (ref. 18) in support of their helix in a disordered protein model, could they comment on the temperature-dependency of this structure? Similar to the R1, R2, and NOE values obtained in ref. 18, the authors could use the RDCs and time scale information of motion to comment on the flexibility or rigidity of their proposed helix and how it may affect its interactions.

We collected a series of HSQC spectra at temperatures from 295 K to 310 K. There were small shifts throughout, but nothing indicative of a structural change or re/unfolding.

- What was the rationale for the authors to shortlist T24, T63, and A87 as specific locations for spin labeling? How do they think changing these sites, for e.g. T29, may impact their findings?

We now explain the rationale for picking these three residues for spin labeling (p. 6). In essence we picked the three sites near the centers of the N-tail, the helix, and the C-tail. We do not think changing the labeling site from T24 to T29 will change our conclusion, in that both T24 and T29 are near the center of the N-tail.

- For ssNMR experiments, why were the calibrated temperature kept different during the experiments, what was the estimated sample temperature in all these cases?

The temperature differences in the SSNMR experiments are generally only related to cooling of the sample when obtaining CP-based measurements. This is because the CP transfer efficiency is hindered by internuclear reorientation and cooling helps with reducing all movement. Typically we try a range of temperatures to ensure there are no structural differences, but then use the temperature that give the best signal. The calibrated temperature is the sample temperature.

What is the stoichiometric ratio of FtsQ and FtsZ proteins identified through co-elution? How does it correlate to the physiological ratios of these proteins in Mtb divisome?

Our NMR titration indicated a 1:1 stoichiometry for FtsQ and FtsZ. The expression levels of FtsQ in *Mtb* are unknown, but recall that FtsZ exists in *Mtb* as polymers (forming the Z-ring), and so not every FtsZ monomer has to bind a FtsQ molecule. Moreover, another protein, SepF, is also known to anchor the Z-ring.

- How do the authors interpret 'motional coupling' to explain the loss of peak intensity for residues 21-45 and 75-84 with the Arg/Ala-rich helix in presence of POPG/POPC vesicles?

We presented such an interpretation in p. 10, second paragraph.

- In the discussion, the authors mention the potential of their findings in drug binding. It would be nice to test a standard drug and tested how their interactions were affected. Are there any similar studies available in the literature? Without any experiments, since it is possible that the drugs do not interact at all, or bind through an entirely different mechanism, currently the discussion seemingly oversells this point.

The inter-subdomain cleft of FtsZ is a known drug-binding site (see ref. 25). Our work is the first to report the binding of FtsQ to FtsZ, potentially to the inter-subdomain cleft, and thus there is yet no study on how drug molecules may interfere with FtsQ-FtsZ binding. We certainly hope our work will now motivate such studies.

- Since the length of the helix is almost thrice a generic helical length, which conditions support stabilizing this specific conformation in the otherwise disordered protein?

We have pointed out that interactions between acidic residues in the N-tail and Arg residues in the helix stabilize this long helix (bottom of p. 6 to top of p.7).

- Although a proposed structure is simulated computationally based on the findings, the authors could include a comment on what may be missing from the experimental data to build a structure for this proposed helix and outline the challenges of experimentally converting this proposed helical wheel into a well-resolved structure. This emphasizes the need for such nicely done comprehensive studies.

The challenge is best illustrated by the complex between FtsQ and acidic membranes. This complex is very dynamic and thus precludes the determination of a single structure by experimental means. However, MD simulations are well suited to characterize molecular details of such fuzzy complexes. The solution and solid-state NMR data provide crucial validation of the MD simulations.

Minor corrections:

- Figure 1 A: Helical region is underlined from 46-74, whereas the helical wheel included 48-72 residues. Why is a discrepancy between experimental and MD data?

We refer to residues 48-72 as the core helical region. The residues at the helical termini are expected to fray or have accumulated “phase shift” so do not align perfectly with the helical wheel.

- Figure 1 B: Peaks assigned to H38 and N85 are not visible in the figure, authors may check the arrows are marked correctly. Is the peak next to A48 not assigned to any residue?

These assignments are correct. H38 and N85 have very low signal intensities in the 2-D HSQC (and hence require very low contour levels for them to show), but show up more clearly when a 3rd dimension is introduced. The intensity assigned to A48 is a single peak.

- The last word in the main text referencing Table S 2, should be ‘non-helical’ instead of nonlocal.

Corrected

- Table S 2: What does superscript 2 refer to (next to nonhelical) ?

“2” is extraneous and removed.

- Fig. S 4: The labels of I10 and A72 are interchanged. In the caption, POPC is misspelled.

Figure corrected and caption corrected.

- I suggest adding an introductory sentence to the results of ssNMR experiments in this study. At present, it seems as though a sentence is missing before this paragraph in the results.

Great suggestion – done (p. 8).

Reviewer #2 (Remarks to the Author):

In this work, the authors used NMR and MD simulation to study the conformation and binding behavior of Arg/Ala-rich region of Mtb FtsQ to its N-terminal region, acidic membrane and GTPase domain of FtsZ. The Arg/Ala-rich helix is adopted, it can effectively bind to GTP domain of FtsZ and affect the formation of Z-ring. Hence, The FtsQ may serve as a membrane anchor of Z-ring. The study about the folding-upon-binding behavior of IDR and the response to various binding partner is a very interesting topic and has apparent biomedical relevance, for example the investigation about the additional binding partner of IDR. With this regards, the binding behavior and conformation of Arg/Ala-rich helix with different partner should be

compared in a more quantitative way, e.g. binding stability, interaction mode and the critical residues for protein interaction. Moreover, the discussion about the mechanism underlying the folding-upon-binding behaviors of this IDR under various environment is recommended.

We have already presented details of the binding modes and critical residues for binding. In particular, Arg residues in the helix participate in interactions with N-terminal acidic residues in solution, with acidic lipids on the membrane surface, and potentially acidic residues in the inter-subdomain cleft of FtsZ. Moreover, the helix is already formed in solution, and thus is not a case of folding-upon-binding.

REVIEWERS' COMMENTS:

Reviewer #1 (Remarks to the Author):

The authors have appropriately addressed my concerns and incorporated suggested edits wherever needed. I accept their provided explanations and modifications, and recommend this manuscript for publication without any further changes.

Final Revision Instructions

To the Author— Please review the editorial comments and requests below and confirm that changes have been made in the manuscript in the right-hand column. **This document must be uploaded** as a related manuscript file.

Please see our final file submission checklist for information about submitting your revised documents.

Files and General Policies	
Main manuscript file must be in Microsoft Word or LaTeX format. LaTeX and Tex article source files must be accompanied by the compiled PDF for reference. The bibliography must be submitted separately (as a .bib file) or contained within the .tex file.	Yes
Each Figure must be provided as a separate file and must be supplied whole, with all panels included in a single document. Figures should be provided at a minimum resolution of 300 dpi at final size. Figure files must only contain images (please also leave out labels such as “Figure 1” etc). Figure captions must instead be included within the main manuscript file, grouped together at the end of the document.	Yes
All figures, tables, and supplementary items must be cited in the manuscript and numbered in the order in which they appear.	Yes
Tables in the main manuscript must be provided in an editable format and should be grouped together at the end of the main manuscript file.	N/A
Please check whether your manuscript contains third-party images, such as figures from the literature, stock photos, clip art or commercial satellite and	N/A

map data. We strongly discourage the use or adaptation of previously published images, but if this is unavoidable, please request the necessary rights documentation to re-use such material from the relevant copyright holders and return this to us when you submit your revised manuscript. An appropriate permissions statement must be present in the relative figure caption for any third-party images.	
Please check that you have not copied any text directly from published work (even your own) without clear attribution, including one or more references. We run a plagiarism detection software and may need to request additional changes if we identify large blocks of identical text.	Checked
An updated editorial policy checklist that verifies compliance with all required editorial policies must be completed and uploaded with the revised manuscript. All points on the policy checklist must be addressed; if needed, please revise your manuscript in response to these points. https://www.nature.com/documents/nr-editorial-policy-checklist.pdf. Please note that this form is a dynamic ‘smart pdf’ and must therefore be downloaded and completed in Adobe Reader. This file will not open in an internet browser.	Yes
The reporting summary will be published alongside your manuscript therefore it needs to accurately represent your work. In this case, please take a closer look at the reporting summary and make sure things are completed correctly. If an item does not apply, for example human participants, I need you to check the NA box next to that item. No section should be left blank. Also, please make sure to include your name and date at the top of the document.	Yes

If you require a new Reporting Summary form, please download it here: https://www.nature.com/documents/nr-reporting-summary.pdf. Please note that this form is a dynamic 'smart pdf' and must therefore be downloaded and completed in Adobe Reader. This file will not open in an internet browser.	
Your paper will be accompanied by a brief editor's summary when it is published on our homepage. Please approve the draft summary below or provide us with a suitably edited version (no more than 250 characters including spaces). The disordered cytoplasmic region of Mtb FtsQ can interact with three partners – the upstream acidic residues in solution, the inner membrane surface, and the GTPase domain of FtsZ – suggesting its role in cell division and as a drug design target.	The disordered cytoplasmic region of Mtb FtsQ can interact with three partners – the upstream acidic residues in solution, the inner membrane surface, and the GTPase domain of FtsZ – suggesting its role in cell division and as a drug target.
ORCID Communications Biology is committed to improving transparency in authorship. As part of our efforts in this direction, we are now requesting that all authors identified as 'corresponding author' create and link their Open Researcher and Contributor Identifier (ORCID) with their account on the Manuscript Tracking System (MTS) prior to acceptance. ORCID helps the scientific community achieve unambiguous attribution of all scholarly contributions. For more information please visit http://www.springernature.com/orcid. For all corresponding authors listed on the manuscript, please follow the instructions in the link below to link your ORCID to your account on our MTS	Yes

before submitting the final version of the manuscript. If you do not yet have an ORCID you will be able to create one in minutes. https://www.springernature.com/gp/researchers/orcid/orcid-for-nature-research IMPORTANT: All authors identified as ‘corresponding author’ on the manuscript must follow these instructions. Non-corresponding authors do not have to link their ORCIDs but are encouraged to do so. Please note that it will not be possible to add/modify ORCIDs at proof. Thus, if they wish to have their ORCID added to the paper they must also follow the above procedure prior to acceptance. To support ORCID's aims, we only allow a single ORCID identifier to be attached to one account. If you have any issues attaching an ORCID identifier to your MTS account, please contact the Platform Support Helpdesk at http://platformsupport.nature.com/	
We regularly highlight papers published in Communications Biology on the journal’s Twitter account (@CommsBio). If you would like us to mention authors, institutions, or lab groups in these tweets, please provide the relevant twitter handles in the right-hand column.	N/A
We would welcome the submission of material for the ‘Featured Image’ section on the Communications Biology home page. Images should relate to the content of your manuscript but need not be contained within the paper. Photographs and aesthetically interesting images are preferred; diagrams are generally not used. Suggestions should be uploaded as a Related Manuscript file. Please provide 1200x675-pixel RGB images. You will also need to submit a completed Image License to Publish. Unfortunately, we cannot promise that your suggestions will be used.	

Supplementary information	
Supplementary Information Format and referencing  ● Supplementary Figures, small Tables, and any supplementary text must be provided in a single PDF. Figures and their captions should be presented together.  ○ If you include a title page, please check that the title and author list matches the main manuscript. ● All Supplementary items must be referred to in the manuscript, and items must be mentioned in numerical order. Please do not include general references to “Supplementary Material”; instead refer to specific items. ● Additional files can be provided as Supplementary Data (Excel files, text files, .zip folders), Supplementary Movies, Supplementary Audio, or Supplementary Software (.zip folder) Supplementary Information files will be uploaded with the published article as they are submitted with the final version of your manuscript. Any highlighting or tracked changes should be removed from the file.	Yes
Supplementary items must be cited in a consistent format. Names of items in the Supplementary file(s) must match those used in the main manuscript. We recommend using the following naming formats: Supplementary Figure 1, Supplementary Table 1, Supplementary Data 1, Supplementary Note 1, and Supplementary References.	Yes

Please provide simulation input files and initial and final coordinate files as Supplementary Data #.	
It's mandatory to provide access to the numerical source data for graphs and charts: We strongly recommend depositing these to suitable repositories (such as Figshare, Dryad, or a data type-specific repository if one exists). Otherwise, all source data underlying the graphs and charts presented in the main figures must be uploaded as Supplementary Data (in Excel or text format). Note that only the data used directly for generating the charts needs to be supplied.	Uploaded as Supplementary Data 1
For any Supplementary Files such as those mentioned above that are not included your combined PDF (e.g. Supplementary Data, Movies, Audio, Software), please provide a title and description for each file here in the column to the right. For example: File name: Supplementary Data 1 Description: The source data behind the graphs in the paper	File name: Supplementary Data 1 Description: The source data behind the graphs in the paper File name: Supplementary Data 2-4 Description: Input, initial and final coordinate files for MD simulations in solution File name: Supplementary Data 5-7 Description: Input, initial and final coordinate files for MD simulations at membrane surface
Title Page	
Please ensure that the author list provided in our manuscript tracking system matches the author list in the main manuscript.	Yes
Please check that your author list and affiliations comply with the following:	corrected

 • Each affiliation must include the institution, city and country. The name of the city must be provided separately from the institution even if it is a part of the institution name; e.g. ‘University of Science and Technology Beijing, Beijing, China’ • Author tagging statements are limited to the following two options: "These authors contributed equally" and "These authors jointly supervised this work", with no more than one of each tag permitted. Please label the equal contribution tag with a symbol. Affiliation #6 is used twice instead of affiliation #5. Please ensure that Affiliation #4 and 5 have their full affiliation.	
Manuscript title Please ensure the title clearly describes the central finding of the paper. We recommend writing the title as a declarative statement of approximately 15 words or fewer. Be sure to include any key species, protein names, or gene names to ensure optimal retrieval of the paper in database searches. Title is approved.	We made a small change to the title to most accurately reflect the central finding.
Main text	
Format of the main text Please ensure your manuscript includes the following sections, presented in this order:  1. “Introduction”: The background and rationale for the work. The final paragraph should be a brief summary of the major results and 	Done

conclusions. The results of the current study must only be discussed in this final paragraph. The Introduction should contain no references to figures or tables. Do not include subheadings. 2. “Results” or “Results and Discussion”. This should be split into subheaded sections; we recommend 1 subheading per main figure or table. Figures should not be embedded in the text but submitted separately. a. Do not use more than 1 layer of subheadings. b. A “Conclusions” paragraph can be included only if the results and discussion are combined into a single section. 3. “Discussion” (optional), without subheadings. 4. Methods, which should be split into subheaded sections. Do not use more than 1 layer of subheadings. To improve readability, we recommend that the main text (Introduction, Results and Discussion) be limited to approximately 5000 words or fewer. Please label the Introduction section.	
Statistical reporting Wherever statistics have been derived (e.g. error bars, box plots, statistical significance) the legend needs to provide and define the n number (i.e. the sample size used to derive statistics) as a precise value (not a range), using the wording “n=X biologically independent samples/animals/independent experiments” etc. as applicable.	N/A
Display items	

Figure captions/legends Figures must have a title that will appear above the Figure and a legend that will appear below the Figure (see e.g. https://www.nature.com/articles/s42003-020-1059-1/figures/1) The Figure title must describe the Figure as a whole and must not contain reference to specific figure panels. The Figure legend must refer to and describe all panels. Abbreviations, symbols, colors, and shading present in the Figure must be defined. Please write out the symbols/colors in words (blue circles, red dashed line, etc.) within these definitions. All figure panels must be labelled using lower case letters. Please refrain from referring to sections of figures as top/bottom/left/right/, etc.	Corrected
Axis and panel labels will be published as received. We recommend using a sans-serif font such as Arial or Helvetica.	Yes
Blots and gels All blots/gels must be accompanied by size markers in every figure panel. Uncropped and unedited blot/gel images must be included as Supplementary Figure(s). The new Supplementary Figure(s) must be cited in the main manuscript text (for example, in the Data Availability Statement). Please pay close attention to our Digital Image Integrity Guidelines and to the following points below:  ● that unprocessed scans are clearly labelled and match the gels and western blots presented in figures. Unprocessed scans must be included in a supplementary figure. 	Included

 ● that control panels for gels and western blots are appropriately described as loading on sample processing controls ● all images in the paper are checked for duplication of panels and for splicing of gel lanes. Finally, please ensure that you retain unprocessed data and metadata files after publication, ideally archiving data in perpetuity, as these may be requested during the peer review and production process or after publication if any issues arise. Please provide the uncropped gel for Figure 4a.	
Tables in the main text Please check that your Tables comply with the following:  ● Do not include shading or colors. All Tables must contain black and white text only. ● Any bold/italic formatting must be either removed or defined clearly in a Table footnote. ● Where Tables contain images, each image should appear in its own cell in the absence of any text. ● All Tables must have a brief title. 	N/A
Methods	
Please ensure that all information present in the Reporting Summary is also in the manuscript. This information is usually most appropriate in the Methods section.	Yes

We allow unlimited space for Methods. The Methods must contain sufficient detail such that the work could be repeated. It is preferable that all key methods be included in the main manuscript, rather than in the Supplementary Information. Please avoid use of “as described previously” or similar, and instead detail the specific methods used with appropriate attribution.	Corrected
The Methods should include a separate section titled “Statistics and Reproducibility” with general information on how the statistical analyses of the data were conducted, and general information on the reproducibility of experiments (also those lacking statistical analysis), including the sample sizes and number of replicates and how replicates were defined.	Done
Please use the Nature templates for NMR, cryo-EM, and X-ray refinement statistics for newly reported macromolecular structures (see https://www.nature.com/commsbio/submit/submission-guidelines#characterisation). These should be presented as Tables in the main manuscript file. Please follow the Nature template for NMR data.	We report chemical shifts in Supplementary Table 1. However, our NMR data were insufficient to determine a structure, due to the disordered and dynamic nature of FtsQ1-99.
Data Policies	
The Data Availability statement must include:  ● Access details for deposited data, including repository name and unique data ID. ● How source data can be obtained. ● A statement that all other data are available from the corresponding author (or other sources, as applicable) on reasonable request. Note 	Yes

that ‘available upon request’ is only appropriate if immediate data access has not been mandated by our policies or by the editors. See here for more information about formatting your Data Availability Statement: http://www.springernature.com/gp/authors/research-data-policy/data-availability-statements/12330880	
Mandatory deposition of raw and processed data is required for:  ● All sequencing data (DNA, RNA, protein) ● Novel human genetic polymorphisms (e.g., dbSNP) ● Linked genotype and phenotype data (e.g., dbGaP for human data) ● GWAS summary statistics or polygenic risk scores ● Novel macromolecular structure ● Gene expression microarray data (must be MIAME compliant) ● Crystallographic data for small molecules ● Mass spectrometry-based proteomics data For more information on mandatory data deposition policies at the Nature Portfolio, please visit http://www.nature.com/authors/policies/availability.html#data For an up-to-date list of approved repositories for each mandatory data type, please visit https://www.springernature.com/gp/authors/research-data-policy/repositories/12327124. Accession code(s) for deposited data must be provided in the Data Availability statement in the final version of the paper. Failure to do so will delay publication. Please ensure data are available prior to publication.	Done: SAXS deposited with ID SASDRA6. We also deposited the NMR chemical shifts to BMRB with ID 51860.

Please deposit SAXS data and provide deposition ID in the Data Availability statement.	
Communications Biology has a strong preference for all data to be deposited in an approved repository. In some cases, data deposition may be required by the editor. We recommend the following data repositories:  ● GenBank (all DNA sequence data) ● NHGRI-EBI GWAS Catalog (GWAS summary statistics) ● PGS Catalog (polygenic risk scores) ● Gene Expression Omnibus (Microarray or RNA sequencing data) ● Sequence Read Archive (WGS or WES data) ● Protein Data Bank (protein structural data) ● OSF (neuroimaging raw data and EEG/EMG/MEG raw data) ● Neurovault (unthresholded statistical maps, parcellations, and atlases produced by MRI and PET studies) ● Image Data Resource (microscopy data) ● PRIDE (proteomics data) Data types without a specific repository can be deposited in a generalist repository, such as figshare or Dryad. For an up-to-date list of approved repositories, please visit https://www.springernature.com/gp/authors/research-data-policy/repositories/12327124.	Done
Data citation Please cite datasets stored in external repositories in the main reference list.	N/A

For previously published datasets, we ask authors to cite both the related research articles and the datasets themselves. For more information on how to cite datasets in submitted manuscripts, please see our data availability statements and data citations policy.	
Code availability Please include a Code Availability statement, indicating whether and how the code can be accessed, including any restrictions to access. In some cases, the editor may require that code be made immediately available. This section should also include information on the versions of any software used, if relevant, and any specific variables or parameters used to generate, test, or process the current dataset. The Code Availability statement must be provided as a separate section after the Data Availability section. Please see our policy on code availability for more information. http://www.nature.com/sdata/for-authors/editorial-and-publishing-policies#code-avail In addition to making the custom code available, please ensure that the version of the code/software described in the paper is deposited in a DOI-minting repository (eg, Zenodo) and that this DOI is also cited in the main Reference list. Please provide simulation input and initial and final coordinate files as Supplementary Data # files and cite in the Code Availability statement.	Done (cited as Supplemented Data 2-7).
End Notes	

Please check that your bibliography complies with the following: ● Your bibliography should start with the heading “References”. The references must be numbered in the order of appearance in the text, then tables, then figures.● Any in-text citations to references (e.g. "Gupta et al. show...") should be followed by their corresponding reference citation number from the reference list.● Manuscript citations must include journal title, article title, volume number, page or article number or DOI, and year of publication.● No publication can be present more than once in the reference list.● No footnotes are permitted in the references or elsewhere. Text should be incorporated into the main text, the Methods section, or the Supplementary Information instead.● Websites should only be listed in the references if they are in common use or curated.● Where possible, preprints in the reference list should be updated with details of the published, peer-reviewed paper.● Citations should be formatted in the text using superscript numbers.	Yes
Please provide a 'Competing interests' statement using one of the following standard sentences: ● The authors declare the following competing interests: [specify competing interests]● The authors declare no competing interests. See our competing interests policy for further information: https://www.nature.com/nature-research/editorial-policies/competing-interests	Done

Please replace the Competing financial interests statement with the standard Competing interests statement.	
Please check that your 'Author Contributions' section individually lists the specific contribution of each author to the work. Each author must be referred to by name or initials. Where multiple authors possess identical initials, they must be clearly disambiguated from one another. See our author contributions policy for further information: https://www.nature.com/nature-research/editorial-policies/authorship#author-contribution-statements	Yes